# Cell-Free DNA as a Surveillance Tool for Hepatocellular Carcinoma Patients after Liver Transplant

**DOI:** 10.3390/cancers15123165

**Published:** 2023-06-13

**Authors:** Joao Manzi, Camilla O. Hoff, Raphaella Ferreira, Renata Glehn-Ponsirenas, Gennaro Selvaggi, Akin Tekin, Christopher B. O’Brien, Lynn Feun, Rodrigo Vianna, Phillipe Abreu

**Affiliations:** 1School of Medicine, University of Sao Paulo, Sao Paulo 05508-900, Brazil; joao.manzi@fm.usp.br (J.M.); camilla.hoff@fm.usp.br (C.O.H.); 2Miami Transplant Institute, Jackson Memorial Hospital, University of Miami, Miami, FL 33136, USA; raphaelladpferreira@gmail.com (R.F.); gselvaggi@med.miami.edu (G.S.); atekin@med.miami.edu (A.T.); cobrien@med.miami.edu (C.B.O.); lfeun@med.miami.edu (L.F.); r.vianna@med.miami.edu (R.V.); 3Medical Affairs Department, CareDx, Inc., Brisbane, CA 94005, USA; rponsirenas@caredx.com

**Keywords:** liver transplant, oncology, hepatocellular carcinoma, cell-free DNA

## Abstract

**Simple Summary:**

The liver is the world’s sixth most common primary tumor site, accountable for nearly 5% of all cancers and over 8% of cancer-related deaths. Hepatocellular carcinoma (HCC) is the predominant type of liver cancer, accounting for about 75% of all primary liver tumors. One of the major therapeutic tools for this disease is liver transplantation. This therapeutic modality, as with the others, faces the obstacle of tumor recurrence, in addition to graft rejection. In this context, cell-free DNA is presented as a new tool for decision-making. In this article, we summarize the main aspects of this new tool, exploring its strengths and weaknesses in the treatment of HCC.

**Abstract:**

The liver is the world’s sixth most common primary tumor site, responsible for approximately 5% of all cancers and over 8% of cancer-related deaths. Hepatocellular carcinoma (HCC) is the predominant type of liver cancer, accounting for approximately 75% of all primary liver tumors. A major therapeutic tool for this disease is liver transplantation. Two of the most significant issues in treating HCC are tumor recurrence and graft rejection. Currently, the detection and monitoring of HCC recurrence and graft rejection mainly consist of imaging methods, tissue biopsies, and alpha-fetoprotein (AFP) follow-up. However, they have limited accuracy and precision. One of the many possible components of cfDNA is circulating tumor DNA (ctDNA), which is cfDNA derived from tumor cells. Another important component in transplantation is donor-derived cfDNA (dd-cfDNA), derived from donor tissue. All the components of cfDNA can be analyzed in blood samples as liquid biopsies. These can play a role in determining prognosis, tumor recurrence, and graft rejection, assisting in an overall manner in clinical decision-making in the treatment of HCC.

## 1. Introduction

The liver is the world’s sixth most common primary tumor site, responsible for approximately 5% of all cancers and over 8% of cancer-related deaths [1]. Both incidence and mortality are expected to rise due to the increase in obesity and metabolic syndrome in the population, despite greater global control of the Hepatitis B and C viruses (HBV and HCV, respectively) through vaccination and treatment [2].

Hepatocellular carcinoma (HCC) is the predominant type of liver cancer, accounting for approximately 75% of all primary liver tumors [3]. Approximately 85% of HCCs occur in low- or middle-income countries, particularly in East Asia and sub-Saharan Africa [4]. The incidence of HCC is closely linked to its risk factors [5], with 80% of HCCs linked to HBV or HCV [6]. In developed countries, nonalcoholic fatty liver disease (NAFLD) is a significant risk factor for HCC [7,8], accounting for 10–20% of HCC cases in the United States (US) [9]. Diabetes and the consumption of alcohol and aflatoxins are also important risk factors for HCC [9,10].

The therapeutic choice for HCC involves several factors, such as patient and tumor characteristics, especially tumor staging, and the availability of each option [11]. To determine the best choices of therapeutic regimen and to establish prognoses, more than a dozen staging systems have been developed [12]. Of these, the most commonly used in clinical practice is the Barcelona Clinic liver cancer (BCLC) staging system (Figure 1), which classifies HCC into stages 0, A, B, C, and D, which are further divided into more specific categories within each group [12,13,14]. In this classification, patients in stage 0 or A, also known as very early and early stage, respectively, are treated from a curative perspective, commonly with liver transplantation (LT) or other locoregional therapies, such as resection and ablative therapies [15,16]. Patients classified as B may also be referred to transplantation when using “extended liver-transplant criteria” according to the institution’s criteria, usually related to minor increases in the number or size of the tumors [14,17,18,19], but are more commonly directed towards other treatments, such as transarterial chemoembolization (TACE) and Y90 radioembolization, while those with more advanced diseases usually receive systemic treatment from a palliative perspective [14].

Liver transplantation is a curative-intent therapeutic method for selected patients with HCC, as it removes both the tumor and the underlying liver disease, with a 5-year survival rate close to 70% [20]. In 1996, Mazzaferro et al. suggested the Milan criteria—one nodule ≤ 5 cm or ≤3 lesions, none > 3 cm, necessarily without metastasis, lymph-node involvement, or significant vascular invasion—for the selection of patients with HCC for transplantation [21]. However, the Milan criteria are highly selective and restrictive, leading to the exclusion of a large number of patients [22,23,24]. Therefore, over time, several other, more comprehensive criteria have been proposed [22,23,24].

However, the Milan criteria remain the most commonly used for selecting these patients, with no other globally accepted criteria for liver transplantation in HCC [25]. Due to the inherent limitations of the Milan criteria, individual institutions and specific governmental organizations have created adaptations, taking into consideration regional specificities, such as the availability of organs, the system of allocation, and transplant waiting times [26].

Due to the often long waiting times until the LT is performed, it is recommended that most patients undergo some sort of locoregional bridging therapy, especially if the anticipated wait time is longer than six months, to avoid disease progression or even exclusion from the list [25]. Many of these treatments, including TACE and radioembolization, can also be used as downstaging therapies to make patients eligible for LT according to the selected criteria [27,28,29].

One of the noteworthy problems in the treatment of HCC is tumor recurrence [10]. Immunosuppression therapies, which are necessary to reduce the risk of organ rejection post-transplantation contribute to the increased risk of tumor regrowth. The tumor recurrence rate after LT is estimated to be 8–20% [26,30,31]. When more permissive criteria are used to select patients for LT, the rate of recurrence increases [32]. Tumor recurrence is most often due to the occult growth of pre-existing metastases and is mainly extra-hepatic, involving the lungs, adrenal glands, bones, peritoneum, lymph nodes, and, more rarely, the brain [33]. In patients with a recurrence of HCC after LT, the median survival is two years [34], with a mean time to recurrence of 13.3 months [35]. However, rigorous surveillance after HCC LT can provide better post-recurrence survival (hazard ratio (HR), 0.94; 95% confidence interval (CI) 0.91–0.98; *p* < 0.01) [35]. 

Currently, the detection and monitoring of HCC recurrence and rejection mainly consist of imaging methods, tissue biopsies, and serum alpha-fetoprotein (AFP) follow-up [36,37]. Currently, computed tomography (CT) and magnetic resonance imaging (MRI) are the most commonly used imaging methods for the diagnosis and follow-up of HCC [37,38]. Among these, MRI has a greater sensitivity than CT (80% vs. 60%, *p* = 0.0023) [39] but also presents significant limitations for lesions smaller than 20 mm (61.7%) [40]; these limitations are even more significant for lesions smaller than 10 mm (40%) [39]. In these uncertain cases, a liver biopsy is an important option, but often it can be unfeasible, such as in many cases in section eight of the liver, and with a high rate of false negatives (30%) and associated risks, such as bleeding and seeding [41,42]. As a biomarker of recurrence, AFP can also play a role, but it has very low specificity, compromising its precision [43].

The development of a surveillance method that can detect HCC recurrence early on, therefore, is an urgent need, which would enable rapid intervention, when necessary [44]. In this context, liquid biopsy presents itself as a new and relevant option, not only for the surveillance of HCC recurrence in LT, but also for the evaluation of LT rejection, allowing better and more precise care for these patients [36,45].

## 2. The Role of Liquid Biopsy and Cell-Free DNA (cfDNA) in Patients with HCC Undergoing Liver Transplantation

### 2.1. Liquid Biopsy

Liquid biopsy is a minimally invasive method for the analysis of non-solid biological material for the detection of cell-derived markers [46]. The origin of liquid biopsy can be traced back to the first report of circulating cell-free DNA (cfDNA), in 1948, by Mandel and Metáis, who isolated DNA from the blood of healthy patients [47]. Nearly 20 years later, in 1966, Tan et al. first observed the correlation between elevated levels of circulating cell-free DNA (cfDNA) and disease, specifically systemic lupus erythematosus [48]. In 1977, the first mention of a correlation between cfDNA and oncologic disease was made, when higher levels of cfDNA were noted in the blood of cancer patients [49]. Although such patients were known to have cfDNA, it took several decades for the technologies necessary to detect and analyze cfDNA to develop [50]. Low levels of cfDNA in the blood and the fragmentation of cfDNA due to its apoptotic origin are just two examples of the obstacles researchers faced in developing methods to systematically employ cfDNA in the diagnosis and monitoring of disease [50].

The refinement of amplification and sequencing methods, however, has led to cfDNA becoming the backbone of the liquid-biopsy movement [50]. When it was first formulated, in 2010, liquid biopsy involved the use of whole circulating tumor cells (CTCs); however, this was rapidly extended to include circulating tumor DNA (ctDNA) and other tumor-derived products, such as extracellular vesicles, tumor-educated platelets, and circulating cell-free RNA [51]. It is currently a field of great interest and activity, with 13,071 articles published with the critical specific terms “CTC”, “ctDNA”, or “cfDNA” between 2018 and 2022, compared to 5124 in the previous five years, according to PubMed [52], representing a growth of more than 150%.

One of the great advantages of liquid biopsy is the plethora of different patient-derived samples on which it can be. Analyses can be performed on different body fluids, such as cerebrospinal fluid (CSF), saliva, ascitic fluid, and, most commonly, blood [53]. The choice of biological material to be analyzed depends on the investigation carried out, such as urine analysis for urinary-tract cancers and CSF for central-nervous-system tumors [54]. Regardless of which biological fluid is chosen, the aim is to evaluate the presence of circulating tumor cells (CTCs), circulating extracellular nucleic acids, such as those found in circulating tumor DNA (ctDNA), microRNA (miRNA), extracellular vesicles, nucleosomes, and various glycoproteins and antigens, such as CA-125, CA-19-9, CEA, and many others [46]. In the context of the present tumor surveillance in oncology, liquid biopsy mainly involves ctDNA, miRNAs, CTCs, and exosomes, which allow the detection and follow-up of neoplastic lesions [55].

When neoplastic or healthy cells move through the process of cell death, regardless of whether it is immune-mediated by cytotoxic cells, they release their DNA into their microenvironment, producing cell-free DNA (cfDNA) [56]. Therefore, as a whole, cfDNA in bodily fluids can be derived from healthy or neoplastic tissues. Circulating tumor DNA (ctDNA) is the subset of cfDNA from tumor cells [57]. Significantly, ctDNA presents characteristics similar to those of its tumor of origin, making it possible to evaluate tumor masses using these free, easily accessible molecules [58].

Although ctDNA is mixed with other cfDNA, its isolation is possible, since ctDNA molecules are smaller than nonmutant cfDNA (Figure 2) [59]. The cut-off value used is 167 base pairs (bp), which is the usual length of DNA wrapped around the chromatosome in nonmutant cells [60]. In tumor cells, ctDNA is typically less than 167 base pairs (bp), with average sizes between 90 bp and150 bp. Therefore, the selection of fragments smaller than 167 bp isolates ctDNA [60]. Additionally, ctDNA also shows identifiable molecular alterations associated with malignancy, such as single-nucleotide mutations, methylations, and cancer-derived viral sequences, which can all also be used as identifying factors [61,62,63].

Furthermore, cfDNA can be analyzed quantitatively and qualitatively. The quantitative analysis of cfDNA molecules, including both ctDNA and non-ctDNA, has prognostic value prior to treatment, can be used to monitor therapeutic response, and can be used in the surveillance of recurrence in several cancers [44,64,65]. The qualitative analysis of ctDNA, on the other hand, can provide critical genetic details on tumor genotypes, such as information about specific mutations and the overall tumor mutational burden (TMB) [44,66]. One important characteristic of ctDNA is its very short half-life, which has been shown to be less than 2 h [67]. Therefore, by analyzing ctDNA, medical professionals gain an extremely up-to-date snapshot of the present oncological situation of the patient [67]. This provides an opportunity for ctDNA to be used for the longitudinal monitoring of disease progression and characteristics, allowing rapid intervention when necessary [68].

### 2.2. Quantitative

The quantitative analysis of cell-free DNA (cfDNA) and circulating tumor DNA (ctDNA) involves the assessment of the absolute and relative abundance levels of these fragments in the blood [69]. A quantitative analysis of cell-free DNA and circulating tumor DNA (ctDNA) has been widely used in other neoplasms for monitoring patients after treatment with curative intent to determine recurrence, residual disease, and, in cases involving transplants, transplant rejection [36]. Studies involving patients with different types of tumor have shown the clinical validity of ctDNA in detecting minimal residual disease 2.7–11 months prior to clinical or radiographic signs of recurrence [70,71,72].

In pancreatic cancer, for example, detectable levels of ctDNA after surgery are correlated with disease-free survival times of less than 10 months [73], making it possible to use the presence of ctDNA as a method for detecting recurrence earlier than conventional imaging-based follow-up (3.1 vs. 9.6 months, *p* = 0.0004) [73]. In urothelial cancer, the addition of atezolizumab in patients with detectable ctDNA in the postoperative period was associated with an increase in disease-free survival (HR = 0.58, 95% CI 0.43–0.79, *p* = 0.0024) and overall survival (0.59, 95% CI 0.41–0.86) [74]. In esophageal squamous cell carcinoma, positive ctDNA after surgery was correlated with a 1-year recurrence-free survival rate of 0%, while the rate for non-detectable patients was 90% (*p* = 0.0008) [75].

Although, in other neoplasms, the use of ctDNA is already more consolidated, in HCC, its use is still gaining ground [76]. Patients with HCC naturally have a cfDNA concentration 3–4 times higher than those of chronic hepatitis patients (53) and almost 20 times those of healthy patients [77,78,79,80]. In patients with HCC, higher concentrations of cell-free DNA are associated with lower overall survival, higher tumor grade, larger tumors, recurrence, and distant metastases [77,78,79,81,82].

In a study with 74 HCC patients specifically focused on liver transplantation as a treatment, Huang et al. monitored ctDNA values pre and post-LT [83]. In the patients with detectable ctDNA following LT, the recurrence rate was 31.7%; in the patients with undetectable ctDNA following LT, the recurrence rate was 11.5% [83]. The same study found a correlation between preoperative ctDNA and recurrence-free survival, with higher values corresponding to shorter time intervals until recurrence (HR, 3.25; CI 95% 1.18–8.97; *p* = 0.01) [83]. In a study of 46 patients, the detection of cfDNA in 100 μL of serum samples before hepatectomy or liver transplantation was positively correlated with an increased incidence of extrahepatic disease (*p* = 0.0102) relative to the group in whose samples cfDNA was not detected (*p* = 0.0386) [84]. Another study, performed on ten patients, found a complete correlation between ctDNA detection and the recurrence of HCC [85]. Jiang et al., in a study with 45 HCC patients, found the same correlation, with an increased recurrence rate for patients with detectable ctDNA (48.6% vs. 0%) [86]. 

Long et al. conducted a study to evaluate the use of cell-free DNA (cfDNA) levels as markers of recurrence post-transplant or hepatectomy, using a cutoff concentration of 2.95 ng/μL to divide the samples into a high-cfDNA group (>2.95 ng/μL) and a low-cfDNA group (≤2.95 ng/μL) [87]. By evaluating the blood samples from 82 patients with hepatocellular carcinoma (HCC) using these cutoffs, it was found that the 28 patients in the high-cfDNA group had a median recurrence time of 14 months, while the patients in the low-cfDNA group had a median recurrence-free time of 19.5 months (*p* = 0.023) [87]. Beyond the quantitative levels of cfDNA, this study also identified the number of tumors and the presence of microvascular invasion as independent risk factors for recurrence (*p* < 0.05) [87].

The quantitative analysis of cfDNA has been correlated with a wide array of tumor characteristics. In HCC, there is a positive association between elevated cfDNA levels and other tumor characteristics, which alter the recurrence rate and surgical indications [88]. Wang et al. analyzed 81 patients for ctDNA via digital-droplet PCR (ddPCR), detecting ctDNA in 70.4% before hepatectomy. The presence of detectable ctDNA was correlated with larger tumors (*p* = 0.001), a greater number of lesions (*p* = 0.001), microvascular invasion (*p* < 0.001), advanced BCLC stages (*p* < 0.001), shorter disease-free survival (*p* < 0.001), and shorter overall survival (*p* < 0.001) [88].

Despite the improvements in immunosuppressive therapeutic regimens (IS) in recent decades, almost one-third of liver-transplant recipients experience acute rejection (AR) [89,90,91]. Higher levels of immunosuppressants may contribute to a decrease in the incidence of AR but increase the chance of adverse reactions, such as renal insufficiency, new or recurrent neoplasms, and infections [92,93,94]. The search for an equilibrium between the likelihood of acute rejection and immunosuppression-related adverse events in each patient translates into an ongoing need for tools to assist in clinical and therapeutic decision-making. Currently, however, the results obtained using IS levels and liver-function tests (LFTs) are suboptimal [95]. This need is especially significant given that AR decreases graft and patient survival in LT [90].

In this context, cfDNA is a new and promising option [96]. When the cells in the donor allograft die, and the nucleic acids become fragmented, double-stranded cell-free DNA is released into the blood [97]. This donor-derived cell-free DNA (dd-cfDNA) is typically 120–160 base pairs, contributing to its use through conventional separation methods [96,98]. Despite the presence of circulating dd-cfDNA in the recipient’s blood, even if the graft is stable, it can be used to detect the occurrence of allograft injury once there is a significant increase in the amount of dd-cfDNA released [99,100]. Beck et al. monitored the amount of donor-derived cfDNA (dd-cfDNA) in patients after liver transplantation, finding a mean value of 6.8% dd-cfDNA in patients with stable liver transplantation and of more than 60% in patients with biopsy-proven rejection [99].

Years later, Schutz et al. found similar results in their follow-up of 115 post-LT patients, in which the patients with stable responses to LT had a progressive drop in dd-cfDNA after transplantation [101]. Other studies found that the monitoring dd-cfDNA levels is a method with similar precision in the identification of graft injury to current laboratory tests [102,103]. Additionally, Levitsky et al. used donor-derived cell-free DNA (dd-cfDNA) as a rejection marker in liver transplantation. Through the use of dd-cfDNA, researchers were able to detect early signs of graft injury and rejection, with the classification of patients into three distinct groups: the dd-cfDNA was significantly different for patients with biopsy-proven acute rejection (AR) vs. patients with normal function (Tx) (AUC = 0.95, 5.3% cutoff) and between AR and patients with acute dysfunction plus no rejection (ADNR) (AUC = 0.71, 20.4% cutoff), with accuracy and negative predictive values (NPV) of 87% and 100% (AR vs. Tx) and 66.7% and 87.8% (AR vs. ADNR), respectively [95].

### 2.3. Qualitative

Beyond the quantitative analysis of ctDNA levels, the qualitative ctDNA analysis can further contribute to a more complete understanding of tumor biology, with the characterization of the main genetic alterations in specific HCC tumors [104,105]. When considering HCC as a whole, there is no clearly dominant mutational pathway; however, many of the mutations converge with the same main oncologic pathways, such as the p53-signaling pathway, telomerase maintenance, CTNNB1 (catenin-cadherin-associated protein, beta1), vascular endothelial growth factor (VEGF), and changes in response to oxidative stress [104,105,106,107,108,109]. Bettegowda et al. found that 75% of patients with advanced-stage HCC had at least one detectable and describable somatic mutation in plasma ctDNA [57]. In an analysis of the ctDNA from 121 patients with advanced HCC using targeted ultra-deep sequencing, the TERT promoter was present in 51%, TP53 in 32%, CTNNB1 in 17%, and PTEN in 8%, while other, less frequent mutations were also identified [110]. The qualitative analysis of ctDNA, therefore, has allowed the further characterization of HCC tumors’ mutational profiles.

Efforts have also been made to find more specific mutations identifiable in ctDNA. In 2020, Oversoe et al. performed a ctDNA analysis of 95 patients with HCC and 45 with liver cirrhosis without HCC in search of the TERT C228T mutation, the most prevalent tumor-associated mutation in HCC [111]. This mutation was found in a ctDNA analysis of 44% of patients with HCC and in none of the patients without HCC [111]. Solid biopsies subsequently found it in 68% of patients with HCC [111]. A positive correlation was found between the presence of the TERT mutation detected in plasma and an advanced TNM stage (*p* < 0.0001) and vascular invasion (*p* = 0.005) [111]. When the mutation was identifiable in the plasma ctDNA, there was an association with increased mortality (adjusted HR 2.16; 95% CI 1.20–3.88; *p* = 0.1), an association that was not found on solid biopsy, but that can be used for prognosis through ctDNA [111].

Mutations in the tumor-suppressor gene, TP53, which plays a crucial role in cell-cycle regulation, DNA repair, and apoptosis, are the most common mutations in human neoplasms and are associated with worse prognoses in patients with HCC who undergo liver transplantation [112]. The TP53 mutations are also associated with vascular invasion (*p* = 0.0029) and shorter recurrence-free survival in HCC patients after liver transplantation when associated with an absence of CTNNB1 mutation (*p* = 0.009) [113]. These mutations can be detected through ctDNA before transplantation, serving as predictive tools [114]. Mutations in TP53 can also be identified through ctDNA in patients with HCC after liver transplantation, serving as predictive markers of recurrence [115].

An additional feature of liquid biopsy is its capacity to isolate miRNAs, which are increasingly shown to provide important information about tumor biology and disease progression [116]. Six miRNAs are considered potentially important markers for HCC after liver transplantation: miR-19a, miR-126, miR-886-5p, miR-223, miR-147, and miR-24 [117]. A high rate of the presence of these miRNAs in patients beyond the Milan criteria is correlated with a recurrence-free survival of 0% at 5 years, while a low presence is correlated with a recurrence-free survival of 60% in the same period in patients with HCC after liver transplantation (*p* < 0.001) [117]. They are also independent predictors of overall survival (*p* = 0.02) [117]. More recently, miR-92b has also gained in importance in the diagnosis of early HCC recurrence post-transplant, with a sensitivity and specificity of 85.7% and 86%, respectively (AUC = 0.925, 95% CI 0.866–0.984, *p* < 0.001) [118].

Other multiple mutations can be found in the ctDNA of HCC patients [36]. In an analysis of 107 HCC patients, almost 60% had at least one single-nucleotide variation, with MutL homolog 1 (13%), serine/threonine kinase 11 (13%), phosphatase and TENsin homolog (9%), and catenin beta 1 (4%) the four most frequent sites [119]. Thus, ctDNA can also be used to identify predominant tumor mutations, enabling more precise targeted therapies pre-transplantation, during bridging and/or downstaging, and post-transplantation in patients with HCC LT [44].

As qualitative cfDNA analyses provide the molecular profiling of neoplasms, they are also of immense value in directing appropriate and efficient targeted therapies [120]. This process, known as precision medicine, in which the molecular characteristics of the disease determine the treatment, is increasingly presented as the future of medicine, and it is widely recommended by central protocols [37,121,122,123]. However, as many of these therapies are based on acting on immune cells, liver transplantation is usually seen as an exclusion criterion from trials, since the activation of defense cells can cause allograft rejection and end-stage organ failure [124,125]. The use of immune-checkpoint inhibitors (ICIs), which are targeted therapy types, is correlated with an allograft-rejection rate of 39% in the cases of liver transplants [124]. In addition, the efficiency of ICIs can be affected by the immunosuppression status of transplanted patients who lack competent immune cells [126]. Nevertheless, there is a possibility that these medications will be used in the future, even for transplanted patients, as new studies suggest a possible future solution to this problem through costimulatory blockade with anti-CD154 and CTLA4-immunoglobulin [127].

## 3. Other Components of Interest in Liquid Biopsy [89,90,91,92,93,94,95,99,100,101,102,103]

Liquid-biopsy methodologies can also involve the search for circulating tumor cells (CTCs), which are released into the bloodstream naturally from primary or metastatic tumors [128]. Although CTCs are necessary for metastasis, fewer than 0.01% of CTCs result in metastasis [61].

These intact cells can be detected and isolated for the evaluation of tumor tissues. However, this process is immensely hampered by their low number, especially in early-stage disease, and short half-life (1–2.4 h) [129]. The use of circulating tumor cells for cancer diagnosis was approved by the U.S. Food and Drug Administration (FDA) in 2004, but only for certain tumors, such as those in prostate, breast, and colorectal cancer [130]. The use of CTCs for HCC was evaluated in the meta-analysis by Cui et al. using, at first, 20 different studies, which included a total of 1191 patients [131]. In this study, a specificity of 0.60 (95% CI 0.57–0.64), a sensitivity of 0.95 (95% CI 0.93–0.96), a negative likelihood ratio (NLR) of 0.36 (95% CI 0.28–0.48), a positive likelihood ratio (PLR) 11.64 (95% CI 5.85–23.14), and a diagnostic odds ratio (dOR) of 38.94 (95% CI 18.33–82.75) were found [131]. In another meta-analysis performed by the same group, using 18 studies to assess overall survival, including 1466 patients, a lower OS was found for CTC-positive HCC patients than for CTC-negative patients (HR 2.31; 95% CI 1.55–3.42, *p* < 0.01) [131].

A drawback of using CTCs as diagnostic biomarkers is the possible lack of specificity, since they mainly use pan-cancer markers, such as the epithelial cell-adhesion molecule (EpCAM) and creatine kinase (CK) [132]. In HCC, only 20–35% of patients present tumors with EpCAM specifically, with different values for each marker [133]. Shen et al. found a correlation between high levels of EpCAM + CTCs and lower survival in patients with unresectable HCC tumors after TACE, in addition to a correlation between high pre-surgical levels and a greater likelihood of recurrence [134,135].

Previously considered solely as a means of disposing of cellular waste products, exosomes gained in importance following their recognition as essential parts of intercellular communication and tumorigenesis, and they can also be found in bodily fluids as part of liquid biopsies [136]. In a manner that is significantly different from ctDNA, exosomes are released from living cells and carry information about their parent cells, providing living-cell information, in contrast to ctDNA, which is released only in cellular death [136,137]. These characteristics allow exosomes to show altered physiological or pathological states in their cells of origin [138,139].

The miR-21 exosome is related to the suppression of apoptosis in HCC cells and is upregulated in HCC patients [140]. Exosomal proteins, such as carcinoembryonic antigen (CEA) and Glypican-3 (GPC-3), are also related to HCC, enabling their use to distinguish healthy from affected patients, which can contribute to the diagnosis and follow-up of these patients [141]. In the study by Na Sun et al., a new purification system for specific extracellular vesicles was evaluated with 158 at-risk cirrhotic patients as a new method for the earlier detection of HCC, with an area-under-receiver-operator-characteristic curve of 0.93 (95% CI 0.86–1; sensitivity 94.4%, specificity 88.5%) [142]. In another study, by Nakano et al., the uses of exosomal miR-92b and circulating AFP were compared among patients with LT HCC; it was found that the exosome predicted the early recurrence of HCC with an AUC of 0.925 (95% CI 0.86–0.98; sensitivity 85.7%, specificity 86%), while the AUC of the AFP was 0.612 (95% CI 0.49–0.73; sensitivity 92.9%, specificity 38.5%) [118].

In routine practice, the use of cfDNA also has the important advantage of being minimally invasive, it can be used serially to follow response, recurrence, and disease progression, and it is relatively easy to isolate in samples compared to other biomarkers [143,144]. Despite these advantages and the frequent publishing of new positive data regarding the use of liquid biopsy, this technology still features several caveats [145]. First, most of the published data on the subject are derived from retrospective proof-of-concept studies, and they are yet to be sufficiently validated by other researchers [146,147]. Even this validation presents critical challenges, since there is still a lack of standards, which makes comparisons enormously difficult [146,148]. Therefore, it is crucial to establish a standard protocol to be followed worldwide so that new studies can be carried out in a reliable way, making it possible to verify and compare them.

In addition to these points, there are still significant problems with the inclusion of liquid biopsy in clinical practice, such as its high costs and complex logistics, since the transport, processing, and storage of samples are expensive and delicate [147,149,150]. Even after the partial or total resolution of these problems, the most likely scenario is that liquid biopsy will be included among other methods rather than becoming a complete replacement for them [145,146].

## 4. Conclusions

Hepatocellular carcinoma is a disease with a high incidence and mortality, for which liver transplantation remains an essential curative treatment. Despite significant advances in neoadjuvant therapies, bridging therapies, and post-surgical care, the management of HCC patients remains a major challenge due to the risk of tumor recurrence and graft rejection. The use of cfDNA in patients undergoing this therapeutic measure is a new option for pre-transplantation prognosis, the monitoring of recurrence and rejection, and aiding clinical decision-making. 

## Figures and Tables

**Figure 1 cancers-15-03165-f001:**
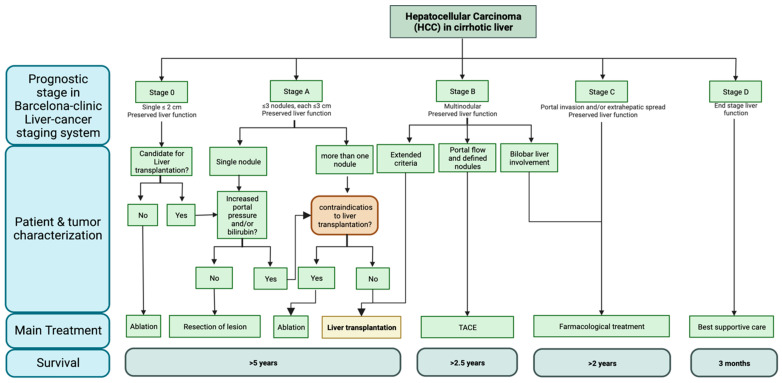
Barcelona Clinic liver cancer (BCLC) staging system, which classifies HCC in stages 0, A, B, C, and D. Each stage suggests specific treatment options and median survival times. Created with BioRender.com (Accessed on 8 December 2022).

**Figure 2 cancers-15-03165-f002:**
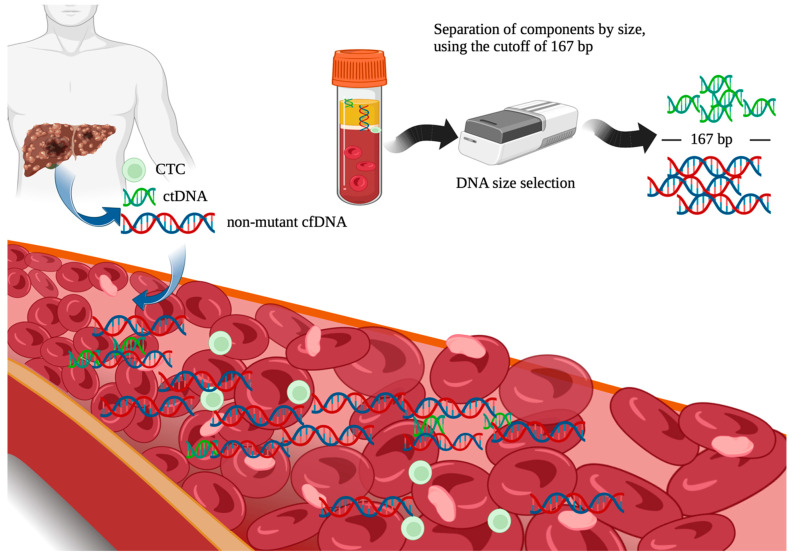
The isolation of ctDNA from other plasma components, such as non-mutant cfDNA, is possible, since ctDNA molecules are smaller than nonmutant cfDNA. The cut-off value used is 167 base pairs (bp). In tumor cells, ctDNA is typically less than 167 base pairs (bp), with average sizes between 90 bp and 150 bp. Therefore, selection of fragments smaller than 167 bp isolates ctDNA. Created with BioRender.com (Accessed on 12 December 2022).

## Data Availability

Not applicable.

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
