# Peer review of "Cell-Free DNA as a Surveillance Tool for Hepatocellular Carcinoma Patients after Liver Transplant"

_cancers, 2023, doi:10.3390/cancers15123165_

Round 1
Reviewer 1 Report
The authors conducted an extensive literature search and wrote this review on the potential use of cfDNA to predict prognosis before liver transplantation, monitor postoperative recurrence and rejection, and aid in clinical decision making in patients with hepatocellular carcinoma.
Hepatocellular carcinoma, a malignant tumor for which organ transplantation improves prognosis, is rather unique.
The fact that cfDNA is excellent for monitoring the dynamics of malignant tumors is not disputed, as it is increasingly reported in other cancers. The usefulness of donor-derived cfDNA in organ transplantation has also been reported, so there is no objection here. However, there is little evidence that this method is useful in hepatocellular carcinoma recurrence, where only 8-20% of post-transplant recurrences occur outside the liver and local therapies such as surgical resection and radiation therapy are still effective. Even if it is useful for other cancer recurrences, the characteristics of recurrence after liver transplantation for hepatocellular carcinoma must be considered. It is needed more discussion on the difference between cfDNA detected with cancer recurrence and donor-derived cfDNA detected with rejection. These should be clearly explained in the abstract. If these points are filled in, the review would be improved to a review that would be of interest to many readers.
Author Response
Reviewer #1 Comments and Response:
- “The fact that cfDNA is excellent for monitoring the dynamics of malignant tumors is not disputed, as it is increasingly reported in other cancers. The usefulness of donor-derived cfDNA in organ transplantation has also been reported, so there is no objection here. However, there is little evidence that this method is useful in hepatocellular carcinoma recurrence, where only 8-20% of post-transplant recurrences occur outside the liver, and local therapies such as surgical resection and radiation therapy are still effective. Even if it is useful for other cancer recurrences, the characteristics of recurrence after liver transplantation for hepatocellular carcinoma must be considered.”
First, we would like to thank you for the comment. We understand and agree with the point raised by the reviewer. We agree that the use of cfDNA for recurrence in hepatocellular carcinoma still needs further exploration and study, and we hope that this review will contribute to strengthening the topic and expanding the knowledge. We have made changes to the text to integrate and better stress the characteristics of the recurrence of hepatocellular carcinoma after liver transplantation into the discussion and the central studies on the recurrence of HCC after liver transplantation. On page 7, in item 3.1, we made changes to better explore the use of cfDNA as a marker of recurrence in HCC after liver transplantation. Together with these are the studies that contribute with evidence for the possible use of this technology in this situation. Thanks again for your contemplation.
- “It is needed more discussion on the difference between cfDNA detected with cancer recurrence and donor-derived cfDNA detected with rejection.”
That was an excellent observation. We understand and agree that further discussion was needed on the differences between cfDNA detected with cancer recurrence and dd-cfDNA detected with rejection. Therefore, we have made meaningful changes to page 10, section 3.3, where we now explore the biological and practical differences between the two entities. The changes made significantly improved the section. Thank you once again.
- “These should be clearly explained in the abstract.”
This is a great suggestion. As recognized and corrected in the previous point, there was not a very clear differentiation between the cfDNA components, notably ctDNA, and dd-cfDNA, and this error was also present in the Abstract. We've made significant changes to the Abstract to clarify component roles.
- “If these points are filled in, the review would be improved to a review that would be of interest to many readers.”
We would like to formally thank you once again for the improvements you suggested. The changes made greatly contributed to an improvement in the overall quality of the review.

Reviewer 2 Report
Dear Authors
I would like to thank you for the opportunity to review this interesting paper focused on a very remarkable and challenging topic that is a lively argument also in daily clinical practice.
Despite the recent technical advances in surveillance, imaging methods, tissue biopsy, and serum alpha-fetoprotein (AFP) still have limited accuracy and precision in monitoring HCC recurrence and graft rejection in transplanted patients. Therefore, the development of new accurate methods that can support clinicians in surveilling this particular fragile group of patients is urgently needed. In this context, liquid biopsy represents a promising new option, not only for the early detection of HCC recurrence but also for the evaluation of transplant rejection.
First of all, the graphical abstract is very beautiful. However, it limits the application of cfDNA only to post-procedural settings. As the Authors correctly stated in the manuscript, pre-procedural analysis of cfDNA could be used to estimate the risk of transplantation failure. Please, revise the figure accordingly.
In the Introduction, “the most used in clinical practice is the Barcelona Clinic Liver Cancer (BCLC) staging system” is correct. However, the Authors should cite and comment the 2022 updated version of BCLC [doi:10.1177/1747493018778713]. In particular, Authors should make a better distinction between the available therapeutic options of patients at very early and early stages, that are the only ones amenable to curative treatments (i.e. resection, transplantation and ablative therapies), and patients at intermediate and advanced stages, that conversely can only be treated with palliative treatments.
Line 60: Please briefly describe some of the most common extended transplantation criteria currently used in clinical practice.
Lines 79-84: I suggest the Authors expand this part and stress the current limitations of imaging surveillance and liver biopsy in monitoring HCC recurrence in transplanted patients. In fact,
Despite the presence of the typical radiologic hallmark—the “wash-in” and “wash-out”— is accurate enough to reach a diagnosis of HCC in nodules larger than 10 mm, the sensitivity of the imaging technique for detecting HCC dramatically drops for liver lesions ranging between 10 and 20 mm [doi: 10.1002/hep.21966], and it is even worse for lesions of less than 10 mm [doi: 10.1148/radiol.14140690]. In these problematic cases or whenever atypical imaging features for HCC are observed, liver biopsy could be a problem-solving technique. Unfortunately, the use of liver biopsy is not always feasible, is not free from risks and is limited by an elevated rate of false negatives (30%), which even increases at a second biopsy (38.9%) [doi:10.1159/000368002][doi:10.3390/jcm11154399]. Please briefly discuss this topic and cite all the aforementioned references.
The section “3. Use of Cell-Free DNA (cfDNA) in patients” should be merged with the previous section. The title of this new section should be changed to “The role of liquid biopsy and Cell-Free DNA (cfDNA) in patients with HCC undergoing liver transplantation” or something similar.
Lines 219-221: In my opinion, it is important to stress the potential role of this new technique in establishing the most efficient and appropriate target therapy, thus dictating patient-specific therapy choices. In fact, in the last years, immunotherapy has led to a major shift in the treatment of HCC and prompted clinical trials; therefore, there is a continuous search for gene markers that can aid in the identification of patients likely to benefit from these new drugs. Organ transplantation has typically been an exclusion criterion in every clinical trial testing immune-checkpoint inhibitors since immunotherapy, through the activation of T cells, can cause allograft rejection and, subsequently, lead to end-stage organ failure in a high percentage of subjects (37.5% and 75%, respectively). Moreover, due to the immunosuppressive status, the efficacy of immunotherapy could be reduced because ICIs require competent T-cell populations to exert their antitumor effects. [doi: 10.3390/ijms24108598] However, treatment with a CTLA-4/Ig fusion protein has been shown to prevent early rejection in an animal model of liver transplantation [doi: 10.1053/jlts.2002.32979], therefore the role of cfDNA in selecting which liver transplant candidates might benefit from a drug rather than another is worthy being investigated in the future. Please discuss the above mentioned topic and cite the suggested references.
The section “3.3 Rejection” should be moved at the end of “3.2 Quantitative” paragraph since the current analysis aims to estimate the rejection rate based on the amount of donor-derived cfDNA.
The section “4. Other uses of liquid biopsy” should be renamed “Other tumor-derived products from liquid biopsy” or something like that. This review, in fact, is mainly focused on cfDNA and not on liquid biopsy.
Finally, please Authors should add a brief paragraph regarding the limitations of liquid biopsy in current clinical practice and how these should be overcome, starting from what they already stated in lines 281-286. For example, it should be acknowledged that the widespread clinical application of liquid biopsy is yet not on the horizon due to both the related cost and technology. Moreover, the majority of data supporting its utility derives from proof-of-concept studies, mainly retrospective, and not validated by different researchers and there needs to be a standardized assay protocol with high sensitivity and specificity. Furthermore, despite the replacement of currently used tools in the management of HCC patients by liquid biopsy biomarkers is unrealistic, they will likely be integrated into the process, providing a stronger predictive power. [doi: 10.3390/cancers13092274.][doi: 10.3390/cancers13040659.].
Finally, I think references should be reformatted as suggested by the Author’s guidelines (Author 1, A.B.; Author 2, C.D. Title of the article. Abbreviated Journal Name Year, Volume, page range)
Author Response
Reviewer #2 Comments and Response:
First, we would like to thank you enormously for the clear, detailed, and careful review. The comments were very accurate, both about opportunities for improvement and recommendations on how to achieve them. Your review was very important for the progress of the article. We hope that we have addressed each of your comments adequately. Thank you so much again.
- “First of all, the graphical abstract is very beautiful. However, it limits the application of cfDNA only to post-procedural settings. As the Authors correctly stated in the manuscript, pre-procedural analysis of cfDNA could be used to estimate the risk of transplantation failure. Please, revise the figure accordingly.”
Thank you very much for the comment. We agree that the graphical abstract was incorrect and gave a wrong idea about the article. We've made changes to correct this error and believe the current one represents the article better.
- “In the Introduction, “the most used in clinical practice is the Barcelona Clinic Liver Cancer (BCLC) staging system” is correct. However, the Authors should cite and comment the 2022 updated version of BCLC [doi:10.1177/1747493018778713]. In particular, Authors should make a better distinction between the available therapeutic options of patients at very early and early stages, that are the only ones amenable to curative treatments (i.e. resection, transplantation and ablative therapies), and patients at intermediate and advanced stages, that conversely can only be treated with palliative treatments.”
Thank you very much for the comment. We agree that there was a need for better exploration and explanation of the BCLC, and we made changes to meet that need. Thank you also for the recommendation regarding the BCLC 2022 update. This source was studied and added to the study. Both of these recommended changes were of great help to the overall quality of the review.
- “Line 60: Please briefly describe some of the most common extended transplantation criteria currently used in clinical practice.”
Thank you for the recommendation. We agree that a brief exploration of what characterizes the "extended criteria" contributes to a better understanding of the text and the importance of the topic since it extends the use of technology to a more significant portion of patients who can undergo transplantation as a therapeutic option. As alterações realizadas podem ser vistas nas atuais linhas 58 - 62.
- Lines 79-84: I suggest the Authors expand this part and stress the current limitations of imaging surveillance and liver biopsy in monitoring HCC recurrence in transplanted patients. In fact,
Despite the presence of the typical radiologic hallmark—the “wash-in” and “wash-out”— is accurate enough to reach a diagnosis of HCC in nodules larger than 10 mm, the sensitivity of the imaging technique for detecting HCC dramatically drops for liver lesions ranging between 10 and 20 mm [doi: 10.1002/hep.21966], and it is even worse for lesions of less than 10 mm [doi: 10.1148/radiol.14140690]. In these problematic cases or whenever atypical imaging features for HCC are observed, liver biopsy could be a problem-solving technique. Unfortunately, the use of liver biopsy is not always feasible, is not free from risks and is limited by an elevated rate of false negatives (30%), which even increases at a second biopsy (38.9%) [doi:10.1159/000368002][doi:10.3390/jcm11154399]. Please briefly discuss this topic and cite all the aforementioned references.
Thank you very much for the thoughtful and accurate comment, not only pointing out the opportunity for improvement but also valuable means to achieve this improvement. The changes made in the indicated place contributed enormously to the introduction and made complete sense with the rest. The suggested sources are of the highest quality and have been carefully studied and included in the text. The marked alterations are in lines 131 - 138 of the third page.
- The section “3. Use of Cell-Free DNA (cfDNA) in patients” should be merged with the previous section. The title of this new section should be changed to “The role of liquid biopsy and Cell-Free DNA (cfDNA) in patients with HCC undergoing liver transplantation” or something similar.
Your suggestion to merge the former section 2 and section 3 has been carefully evaluated and implemented. We agree that the merging of the two sections with the name change contributes to an overall superior understanding of the subject by readers. In order to maintain the internal organization, other changes were made in this new section to facilitate comprehension, such as the creation of subsection 2.1 Liquid Biopsy, which encompasses the entire discussion of this new technology, from biological bases to practical considerations. Thank you once again for the precise recommendation.
- Lines 219-221: In my opinion, it is important to stress the potential role of this new technique in establishing the most efficient and appropriate target therapy, thus dictating patient-specific therapy choices. In fact, in the last years, immunotherapy has led to a major shift in the treatment of HCC and prompted clinical trials; therefore, there is a continuous search for gene markers that can aid in the identification of patients likely to benefit from these new drugs. Organ transplantation has typically been an exclusion criterion in every clinical trial testing immune-checkpoint inhibitors since immunotherapy, through the activation of T cells, can cause allograft rejection and, subsequently, lead to end-stage organ failure in a high percentage of subjects (37.5% and 75%, respectively). Moreover, due to the immunosuppressive status, the efficacy of immunotherapy could be reduced because ICIs require competent T-cell populations to exert their antitumor effects. [doi: 10.3390/ijms24108598] However, treatment with a CTLA-4/Ig fusion protein has been shown to prevent early rejection in an animal model of liver transplantation [doi: 10.1053/jlts.2002.32979], therefore the role of cfDNA in selecting which liver transplant candidates might benefit from a drug rather than another is worthy being investigated in the future. Please discuss the above mentioned topic and cite the suggested references.
Once again, we would like to thank you for the accurate and detailed recommendations. The additional discussion on targeted therapies arising from qualitative cfDNA analysis was a great addition to the rest of the discussion. The references brought to carry out this discussion were also excellent and of great value for the work. The changes made can be seen on page 8, lines 467-478. Many thanks again for the detailed and pertinent recommendations.
- The section “3.3 Rejection” should be moved at the end of “3.2 Quantitative” paragraph since the current analysis aims to estimate the rejection rate based on the amount of donor-derived cfDNA.
We understand and agree with your recommendation.
- The section “4. Other uses of liquid biopsy” should be renamed “Other tumor-derived products from liquid biopsy” or something like that. This review, in fact, is mainly focused on cfDNA and not on liquid biopsy.
Many thanks again for the recommendation. We agree that the previous name, "Other uses of liquid biopsy," does not specify what will be done next. We changed the name to "Other Components of Interest in Liquid Biopsy."
- Finally, please Authors should add a brief paragraph regarding the limitations of liquid biopsy in current clinical practice and how these should be overcome, starting from what they already stated in lines 281-286. For example, it should be acknowledged that the widespread clinical application of liquid biopsy is yet not on the horizon due to both the related cost and technology. Moreover, the majority of data supporting its utility derives from proof-of-concept studies, mainly retrospective, and not validated by different researchers and there needs to be a standardized assay protocol with high sensitivity and specificity. Furthermore, despite the replacement of currently used tools in the management of HCC patients by liquid biopsy biomarkers is unrealistic, they will likely be integrated into the process, providing a stronger predictive power. [doi: 10.3390/cancers13092274.][doi: 10.3390/cancers13040659.].
We agreed that there were great gains in having a more detailed discussion of the limitations of using liquid biopsy. We have studied extensively and used the recommended sources to improve this need. Many thanks again for the detailed and accurate recommendation.
- Finally, I think references should be reformatted as suggested by the Author’s guidelines (Author 1, A.B.; Author 2, C.D. Title of the article. Abbreviated Journal NameYear, Volume, page range)
Thank you very much for the recommendation. We have changed the style of references to the American Medical Association 11th edition (brackets) following the instructions in the Author's guidelines.

Round 2
Reviewer 2 Report
The authors addressed raised points appropriately. However, they did not cite the suggested reference [doi: 10.3390/ijms24108598] in lines 276. After they add this important reference, the manuscript can certainly be accepted.
Kind regards
Author Response
Thank you again for the thoughtful and accurate review. The new reference has been studied and added in the suggested place. Thank you very much for all the assistance.